# Inflammatory Skin Disease Causes Anxiety Symptoms Leading to an Irreversible Course

**DOI:** 10.3390/ijms24065942

**Published:** 2023-03-21

**Authors:** Shohei Iida, Hirotaka Shoji, Fumihiro Kawakita, Takehisa Nakanishi, Yoshiaki Matsushima, Makoto Kondo, Koji Habe, Hidenori Suzuki, Tsuyoshi Miyakawa, Keiichi Yamanaka

**Affiliations:** 1Department of Dermatology, Mie University Graduate School of Medicine, 2-174 Edobashi, Tsu 514-8507, Japan; kmcasters@med.mie-u.ac.jp (S.I.); t-nakanishi@med.mie-u.ac.jp (T.N.); matsushima-y@med.mie-u.ac.jp (Y.M.); kondomak@med.mie-u.ac.jp (M.K.); habe-k@med.mie-u.ac.jp (K.H.); 2Division of Systems Medical Science, Center for Medical Science, Fujita Health University, Toyoake 470-1192, Japan; hshoji@fujita-hu.ac.jp (H.S.); miyakawa@fujita-hu.ac.jp (T.M.); 3Department of Neurosurgery, Mie University Graduate School of Medicine, 2-174 Edobashi, Tsu 514-8507, Japan; fumiwo0219@med.mie-u.ac.jp (F.K.); suzuki02@med.mie-u.ac.jp (H.S.)

**Keywords:** inflammatory skin, mouse model, atopic dermatitis, psoriasis, cytokine, anxiety, JAK inhibitor, Lipocalin 2, S100a8

## Abstract

Intense itching significantly reduces the quality of life, and atopic dermatitis is associated with psychiatric conditions, such as anxiety and depression. Psoriasis, another inflammatory skin disease, is often complicated by psychiatric symptoms, including depression; however, the pathogenesis of these mediating factors is poorly understood. This study used a spontaneous dermatitis mouse model (KCASP1Tg) and evaluated the psychiatric symptoms. We also used Janus kinase (JAK) inhibitors to manage the behaviors. Gene expression analysis and RT-PCR of the cerebral cortex of KCASP1Tg and wild-type (WT) mice were performed to examine differences in mRNA expression. KCASP1Tg mice had lower activity, higher anxiety-like behavior, and abnormal behavior. The mRNA expression of S100a8 and Lipocalin 2 (Lcn2) in the brain regions was higher in KCASP1Tg mice. Furthermore, IL-1β stimulation increased Lcn2 mRNA expression in astrocyte cultures. KCASP1Tg mice had predominantly elevated plasma Lcn2 compared to WT mice, which improved with JAK inhibition, but behavioral abnormalities in KCASP1Tg mice did not improve, despite JAK inhibition. In summary, our data revealed that Lcn2 is closely associated with anxiety symptoms, but the anxiety and depression symptoms caused by chronic skin inflammation may be irreversible. This study demonstrated that active control of skin inflammation is essential for preventing anxiety.

## 1. Introduction

Psoriasis and atopic dermatitis are chronic and intractable inflammatory skin diseases. Intense itching significantly reduces the quality of life (QOL), and atopic dermatitis is often associated with psychosocial distress, which can lead to anxiety and depression [1]. A cohort study with psoriasis identified notable symptoms of depression and anxiety; such a correlation between psoriasis and anxiety and depression essentially leads to psychological sequelae [2]. Comorbid depression and anxiety disorders occur in up to 25% of general practice patients. About 85% of patients with depression have significant anxiety, and 90% of patients with anxiety disorder have depression [3]. To evaluate the psychological symptoms of uncontrolled dermatitis, we conducted an experimental behavioral analysis using a mouse model of spontaneous dermatitis, keratin 14-driven caspase-1 overexpressing (KCASP1Tg) mice. The characteristics, as well as histological and behavioral profiles, fulfill 7 out of the 8 components of the Hanifin and Rajka diagnostic criteria for atopic dermatitis, and KCASP1Tg mice were used as the model for developing atopic dermatitis [3]. Moreover, microarray analysis of the cortex of 24-week-old KCASP1Tg and wild-type (WT) mice were performed and focused mainly on Neutrophil Gelatinase-associated Lipocalin, NGAL (Lcn2), which was highly expressed. We investigated the relationship between psychiatric symptoms and Lcn2 levels in this uncontrolled mouse model. Because the site of cutaneous inflammation is composed of a network of many different inflammatory cytokines, multi-cytokine-targeted drugs, Janus kinase (JAK) inhibitors, were tested as a potential therapeutic option.

## 2. Results

### 2.1. Behavioral Experiments and Janus Kinase (JAK) Inhibitors Dosing Schedule

KCASP1Tg (*n* = 7) and WT (*n* = 10) mice were orally given cerdulatinib or PBS as a control, starting at 6 weeks old, 5 times a week until 18 weeks old when the behavioral experiments were completed. Behavioral analysis was conducted in 4 groups. At 18 weeks old, when the behavioral analysis was completed, all mice were evaluated using the EASI score (head and neck only; full score 7.2), which is mainly used for the evaluation of atopic dermatitis. Eighteen-week-old KCASP1Tg mice had a decrease in body weight and whole brain weight compared to WT mice. The EASI score of KCASP1Tg mice significantly improved after cerdulatinib treatment (Figure 1).

### 2.2. Behavioral Analysis of KCASP1Tg and No Behavioral Changes with JAK Inhibitor Administration

To observe changes in psychiatric symptoms caused by dermatitis and JAK inhibitors, behavioral analysis was performed in 4 groups of 12-week-old KCASP1Tg and WT mice treated with JAK inhibitors and PBS as controls, respectively, according to the schedule shown in Figure 1a. JAK inhibitors or PBS were administered during behavioral analysis. KCASP1Tg mice had no advantage in grip strength, pain sensation, coordinated locomotion or motor learning, working and reference memory, and contextual fear memory compared to WT mice. However, they showed hypoactivity in LD, light/dark transition test; OF, open field test; EP, elevated plus maze test; SI, social interaction test; PS, Porsolt forced swim test; and FZ, contextual and cued fear conditioning test. Moreover, KCASP1Tg mice showed a tendency toward anxiety in LD and OF, decreased social interest in three-chamber test, and reduced startle amplitude in the acoustic startle response test compared to WT mice (Figure 2). Summarizing those behavioral analyses that showed hypoactivity suggested a tendency toward anxiety or decreased social interest and is included in the Appendix A.

### 2.3. Microarray Analysis of Cortex from KCASP1Tg and WT Mice

To evaluate the genes responsible for behavioral changes in the mouse model of long-lasting dermatitis, the cerebral cortices of 24-week-old KCASP1Tg and WT mice underwent microarray analysis. *Erdr1* (erythroid differentiation regulator 1) and *Lcn2* (Neutrophil Gelatinase-associated Lipocalin; NGAL) were upregulated more than 4-fold in KCASP1Tg mice compared to WT mice, and S100a8, whose expression is upregulated in chronic inflammatory diseases, was also upregulated 1.57-fold (Figure 3).

### 2.4. Erdr1, Lcn2, S100a8 mRNA Expression in Cortex, Amygdala, Hippocampus, and Hypothalamus and Effects of JAK Inhibitors

Based on microarray analysis, we examined the differential mRNA expression of Erdr1 and Lcn2 in four brain regions of 18-week-old mice, in which behavioral analysis was completed. We also measured the mRNA level for S100a8, one of the S100 protein family members, which is upregulated in many chronic inflammatory diseases. The cerebral cortex contains motor and somatosensory cortices. The amygdala, which is primarily involved in emotional expression and emotional behavior, the hippocampus, which is involved in intellectual function and memory, and the hypothalamus, which is the general center for endocrine and autonomous function regulation, were each sampled and evaluated using real-time polymerase chain reaction (RT-PCR) analysis. There were no predominant differences in Erdr1 expression between brain regions and between groups. KCASP1Tg mice showed increased mRNA expression of Lcn2 and S100a8 in all regions of the cortex, amygdala, hippocampus, and hypothalamus compared to WT mice. In KCASP1Tg mice, cerdulatinib treatment decreased Lcn2 mRNA expression in the amygdala and hypothalamus, as well as S100a8 mRNA expression in the cortex, amygdala, and hypothalamus (Figure 4).

### 2.5. Cytokine-Stimulated Lcn2 mRNA Expression in Astrocytes and Evaluation of Lcn2 Protein Levels in Plasma

In KCASP1Tg mice, various inflammatory cytokines, including IL-1β, are released from dermatitis skin lesions into the blood. To evaluate how these inflammatory cytokines affect the central nervous system (CNS), we cultured astrocytes, a type of glial cell in the central nervous system, with TNF-α, IL-17A, and IL-1β in vitro to examine the changes in Lcn2 mRNA expression. IL-1β stimulation predominantly increased Lcn2 mRNA expression in astrocytes in vitro. The amount of Lcn2 protein in the plasma of each of the four groups of 18-week-old mice, where behavioral analysis was performed, was also measured. KCASP1Tg mice showed a predominant increase compared with WT mice and a marked decrease after cerdulatinib treatment (Figure 5).

## 3. Discussion

The skin is one of the largest immune organs in the human, and it plays a vital role in maintaining homeostasis in the human body and is a barrier against pathogens. Here, we used KCASP1Tg mice, a dermatitis model, to investigate the association between dermatitis and behavior. First of all, this mouse model of dermatitis reveals spontaneous dermatitis without any external triggers. About 8 weeks after birth, the mice started to show eruption around the eyes and neck. The lesions extended to the ears and legs and finally became generalized. Dermatitis started as erosive erythematous patches and moist crusted areas. After repeated re-epithelization and inflammation, the lesions progressed to chronic lichenified plaques. Almost all the mice developed cataracts, as is the case with AD. The frequency of scratching behavior increased remarkably with the onset of the skin lesions. The skin before the onset of skin symptoms showed no histopathological changes. After onset, it presented remarkable epidermotropic cell infiltration, papillomatous proliferation and thickening of the epidermis, and partial epidermal defects, crusts, and parakeratosis. These changes resembled those observed in the acute stage of human atopic dermatitis. The dermis showed an increase in the number of infiltrating CD4 T cells and a remarkable increase in toluidine blue-positive mast cells, similar to the findings in atopic dermatitis. The serum histamine levels in the model mice increased according to the extension of the skin lesions. Serum histamine levels corresponded with an increase in mast cells in the skin. Serum IgE levels increased by the time of symptom onset and showed remarkable elevation, correlating with the severity of the skin rash. The expression of cytokine mRNAs in the affected skin showed an elevation of IL-4 and IL-5, which are not detected in normal mice. This model satisfies all the essential elements of the internationally accepted diagnostic criteria by Hanifin-Rajka, i.e., (1) itching, (2) typical distribution of skin rash, (3) chronic recurrent dermatitis, and (4) family history. At least three of these major criteria are required. Our model satisfies 11 of 12 minor elements, including xeroderma, elevated serum IgE, early onset, dermatitis of the hands and feet, cheilitis, conjunctivitis, cataract, facial erythema, lesions around the eyes, and aggravation due to environmental changes [4]. As described above, this mouse model of chronic inflammatory dermatitis can be evaluated as type 2 atopic dermatitis.

From the previous studies, skin inflammation affected arteries, and arteriosclerosis was detected not only in the abdominal aorta, but also in the peripheral basilar arteries [5]. These arterial abnormalities were partially ameliorated after administering antibody preparations [5]. Cytokines produced locally in the skin may reach the adipose tissue of the abdomen through the bloodstream, leading to adipocyte denature and subsequent adipocytokine release, which contributes to the systemic inflammatory cascade [6]. Hence, the effects on arteries may be due to direct effects on the vascular endothelium by sustained elevated inflammatory cytokines in the blood [7], as well as the direct effects of adipocytokines from fatty tissues surrounding major blood vessels [6]. Statistics have shown that patients with psoriasis vulgaris, atopic dermatitis, and eczema have a high complication rate of coronary artery and cerebrovascular diseases, which are often fatal [8,9,10,11]. The hardening concept of this vasculature is termed as “inflammatory skin march” [7]. In inflammatory skin conditions, osteoporosis may be complicated by a decrease in the vascular network and an increase in osteoclasts, together with the reduction of osteoblasts [12]. Male infertility is also related to sperm hypoplasia, presumably caused by the direct effect of an increase in inflammatory cytokines from skin lesions [13].

Long-lasting intractable dermatitis may cause depression and anxiety symptoms [1,2]. The protein Lcn2 is related to insulin resistance, obesity, and atherosclerotic diseases. Moreover, it is an inflammatory protein involved in different age-related CNS diseases and risk factors. In dermatitis, serum Lcn2 levels in patients with psoriasis were significantly higher than those in healthy controls [14]; in patients with psoriasis, serum Lcn2 concentrations significantly correlated with the visual analog scale (VAS) [15]. Moreover, Lcn2 may potentiate the psoriasis development by enhancing Th17- and antimicrobial peptide-mediated inflammation [16], and Lcn2 expression is increased in the periphery and brain in other age-related CNS diseases and their risk factors [17]. Studies show that Lcn2 contributes to various neuropathophysiological processes of age-related CNS diseases, including exacerbated neuroinflammation, cell death, and iron dysregulation, which may negatively impact cognitive function [17]. As for the effect of Lcn2 on behavior, mice exposed to prolonged cerebral Lcn2 levels experienced a reduction in spatial reference memory, as indicated by Y-maze assessment [18], and genetic deficiency of caspase-1 decreased depressive- and anxiety-like behaviors and conversely increased locomotor activity and skills [19]. However, Lcn2 may function as a potential protective factor for the central nervous system (CNS) in response to systemic inflammation [20]. We conducted a more detailed behavioral analysis to determine other behavioral changes in the mice with chronic persistent dermatitis. Compared to WT mice, KCASP1Tg mice were less active in several behavioral experiments, including LD, OF, EP, SI, PS, and FZ, suggesting anxiety-like behavior in LD and OF and decreased social interest in CSI. On the other hand, there were no significant differences between the two groups in cooperative movement, motor learning, working memory, and reference memory.

Recently, a multi-cytokine-targeted drug, JAK inhibitor, was used to treat psoriatic arthritis and atopic dermatitis with a good response. Here, we examined the effects of JAK inhibitors on anxiety symptoms in a mouse using JAK inhibitors. Cerdulatinib hydrochloride is an oral, active multi-target tyrosine kinase inhibitor, especially for JAK1, JAK2, JAK3, and TYK2. Skin eruptions were ameliorated in JAK-treated mice, but psychiatric symptoms, including anxiety, showed no improvement. In contrast, mRNA expression of Lcn2 and S100a8, inflammatory proteins, especially in the amygdala and hypothalamus of KCASP1Tg mice brains, was significantly improved by cerdulatinib treatment, similar to plasma Lcn2 expression.

Astrocytes are glial cells in the CNS that not only support the structural network of the CNS, but also regulate various conditions around astrocytes through mass transport. KCASP1Tg mice are mouse models of chronic dermatitis where human caspase-1 is linked to the keratin 14 promoters. Keratin 14 is expressed in the basal cells of the epidermis, and caspase-1 cleaves and matures precursors of IL-1β and IL-18, increasing IL-1β and other inflammatory cytokines in the blood [21,22,23]. In vitro, astrocytes showed predominant expression of Lcn2 mRNA over other inflammatory cytokines upon IL-1β stimulation; in KCASP1Tg mice, in which inflammatory cytokines, including IL-1β, circulate in the blood, Lcn2 expression is elevated in the brain.

These findings suggest that Lcn2, an inflammatory protein, may influence hypoactivity and anxiety symptoms. In this study, although the skin symptoms in KCASP1Tg mice improved after treatment with cerdulatinib, psychiatric symptoms, including anxiety, did not improve, suggesting the possibility of irreversible changes. Proactive control of dermatitis and the prevention of anxiety and depressive symptoms, before daily life is compromised, is advisable.

## 4. Materials and Methods

### 4.1. Animals

In a mouse model of spontaneous dermatitis, keratin 14-driven caspase-1 overexpressing (KCASP1Tg) mice started to show the first symptoms of dermatitis at approximately eight weeks of age, and cutaneous inflammation spread from the face to the whole body [21,22,23]. Six-week-old female KCASP1Tg mice and C57BL/6N littermate (WT) mice were used. Mice were housed in an environmentally conditioned room at 21 ± 2 °C with a 12:12 h light cycle, 60% humidity, and food and water available ad libitum. The Mie University Board Committee approved the experimental protocol for Animal Care and Use (No. 22-39-5-1). Behavioral analysis was conducted at Fujita Health University. After arriving at Fujita Health University animal facility, mice were group-housed (four per cage) in plastic cages (227 × 323 × 127 mm) with paper chips for bedding (Paper Clean; Japan SLC, Inc., Shizuoka, Japan). Rooms were maintained under a 12 h light/dark cycle (lights on at 7:00 A.M.) at 23 ± 2 °C. Animals had free access to food (CRF-1; Oriental Yeast Co., Ltd., Tokyo, Japan) and filtered water throughout the experiments. All experimental procedures were started one week or more after arrival. The Institutional Animal Care and Use Committee of Fujita Health University approved all courses.

### 4.2. Blood Sampling

After behavioral experiments, all mice were euthanized by cervical dislocation. Blood was sampled via cardiac puncture, placed in a 1.5 mL tube containing heparin, and centrifuged (6000 rpm for 5 min) to separate the plasma. The collected plasma was stored at −80 °C until examination.

### 4.3. Microarray Analysis

Total RNA was extracted from the cortex using the Tri reagent (Molecular Research Center, Cincinnati, OH, USA). After euthanasia, we rapidly sampled the cortex of 24-week-old KCASP1Tg mice (n = 5) and WT mice (n = 5). RNA was isolated using chloroform (Nacalai Tesque, Kyoto, Japan) and precipitated using isopropanol (Nacalai Tesque). The RNA concentration was measured using a NanoDrop Lite spectrophotometer (Thermo Fisher Scientific, Worsham, MA, USA). Extracted RNA was transported to KURABO INDUSTRIES LTD (Osaka, Japan) for microarray analysis. The results were analyzed using ”Transcriptome Viewer”. Heat maps and clustering of significant miRNAs are presented for the microarray experiments.

### 4.4. Oral Administration of JAK Inhibitors

Six-week-old female KCASP1Tg and WT littermate mice were treated with active multi-target tyrosine kinase inhibitors, specifically JAK1, JAK2, JAK3, and TYK2, and cerdulatinib hydrochloride (Astellas, Tokyo, Japan), dissolved in corn oil (Fujifilm, Osaka, Japan). PBS (Fujifilm) was administered orally as a control. The treatment involved administering 5 mg/kg cerdulatinib five times a week until 18 weeks of age when the behavioral experiments were completed [24]. All mice were sacrificed at 18 weeks old, and their brains were analyzed.

### 4.5. Behavioral Analysis

KCASP1Tg (n = 15) and WT mice (n = 20) were delivered to Fujita Health University where behavioral analysis was conducted, according to the schedule in Figure 1a. One KCASP1Tg mouse (in PBS) was found dead three day after arrival and was excluded from the behavioral analysis. In the tail suspension test, one KCASP1Tg mouse was excluded because its tail had been damaged by dermatitis.

#### 4.5.1. General Health and Neurological Screen

General health and neurological screening were conducted for two days. On day 1, physical characteristics and neurological reflexes (rectal temperature, body weight, righting reflex, whisker twitch reflex, ear twitch reflex, and visual placing reflex, a forepaw extension when lowered toward a visible surface) were assessed. To measure neuromuscular strength, mice were placed on a wire mesh that was then inverted, and the latency to fall from the mesh was recorded with a 60-s cutoff time. On day 2, forelimb grip strength was measured by holding the mice by their tails and lifting them so that their forepaws could grasp the wire grid of a grip strength meter (O’Hara & Co., Tokyo, Japan). The mice were gently pulled backward by their tail until they released the grid. The peak force applied by the forelimbs was recorded in newtons (N). Each mouse was tested thrice, and the highest value was used for statistical analysis.

#### 4.5.2. Light/Dark Transition Test

The light/dark transition test, initially developed by Crawley et al. [25], was performed to assess anxiety-like behavior, as previously described [26]. The apparatus consisted of a cage (21 × 42 × 25 cm) divided into two sections of equal size using a partition with a door (O’Hara & Co.). One chamber had white plastic walls and was brightly illuminated (390 lux) by lights attached above the chamber ceiling. The other chamber had dark black plastic walls (2 lux). Both chambers contained white plastic floors. Mice were placed in the dark chamber and allowed to move freely between the two chambers for 10 min, with the door open. The distance traveled (cm), the total number of transitions, latency to first enter the light chamber(s), and time spent in the light chamber(s) were recorded automatically using the ImageLD program (see “Section 4.5.14”).

#### 4.5.3. Open Field Test

The open field test was performed to assess locomotor activity and anxiety-like behavior [27] in the open field apparatus using the VersaMax Animal Activity Monitoring System (40 × 40 × 30 cm; Accuscan Instruments, Columbus, OH, USA), in which the center area was illuminated to 100 lux by lights attached above the ceiling. The central area was defined as 20 cm × 20 cm. Each mouse was placed in one corner of an open field. The total distance traveled (cm), vertical activity (measured by counting the number of photobeam interruptions), and time spent in the central area(s) was recorded. Stereotypic counts (beam-break counts for stereotyped behaviors) were automatically recorded using the activity monitoring system, for the entire 30 min period, after mice were placed in the apparatus. Behavioral data were analyzed in 5 min blocks.

#### 4.5.4. Elevated Plus Maze Test

The elevated plus maze test, widely used to assess anxiety-like behavior [28], was performed, as previously described [29]. The apparatus consisted of two open arms (25 × 5 cm) and two enclosed arms of the same size, with 15-cm-high transparent walls and a central square (5 × 5 cm) connecting the arms (O’Hara & Co.). The arms and central square floors were made of white plastic plates and elevated 55 cm above the floor. To prevent mice from falling off the open arms, the arms were surrounded by a raised ledge (3 mm thick and 3 mm high, transparent plastic). Arms of the same type were located opposite each other. The illumination level of the central area was 100 lux. Each mouse was placed in the central square of the maze, facing one of the closed arms. The number of arm entries, distance traveled (cm), percentage of entries into the open arms, and percentage of time spent in the open arms were measured during a 10 min test period. Data acquisition and analysis were performed automatically using ImageEP software (see “Section 4.5.14”).

#### 4.5.5. Hot Plate Test

The hot plate test was performed to evaluate sensitivity to painful stimulus. Mice were placed on a hot plate maintained at 55.0 ± 0.2 °C (Columbus Instruments, Columbus, OH, USA). The latency to the first paw response(s) was recorded with a 15 s cutoff time. A paw response was defined as either a foot shake or paw lick.

#### 4.5.6. Social Interaction Test

A social interaction test was conducted to measure social behavior in a novel environment. Pairs of weight-matched mice of the same treatment group, housed in different cages, were placed in a white plastic box (40 cm × 40 cm × 30 cm) and allowed to explore freely for 10 min. Mice were recorded using a video camera placed above the box. Images were captured at three frames per second and transferred to a computer. The distance traveled by each mouse between two successive frames was automatically calculated using ImageSI software (see “Section 4.5.14”). The total number of contacts, total duration of contacts (s), total duration of active contacts (s), mean duration per contact (s), and total distance traveled (cm) were recorded using the software. Active contact was measured when two mice contacted each other and one or both moved with a velocity of at least ten cm/s.

#### 4.5.7. Rotarod Test

Mice were placed on rotating drums (3 cm diameter) of an accelerating rotarod (UGO Basile, Comerio, Italy) to evaluate motor coordination and balance. The speed of the rotarod was increased from 5 to 40 rpm over 5 min. The latency to fall off the rotating rod was recorded with a 5 min cutoff time for 3 daily trials over 2 consecutive days.

#### 4.5.8. Three-Chamber Test

A three-chamber social approach test was conducted to assess sociability and social novelty preference [30]. The testing apparatus consisted of a rectangular three-chambered box and lid with a video camera (O’Hara & Co.). The dividing walls of the chamber were made of transparent plastic, with small square openings (5 × 3 cm), allowing access to each chamber (20 × 40 × 47 cm). A small round wire cage (9 cm in diameter, 11 cm in height, with vertical bars 0.5 cm apart) was located in the corner of the left and right chambers. Test mice were first placed in the middle chamber and allowed to explore the entire test chamber for 10 min. Immediately after 10 min, test mice were placed in a clean holding cage and a male C57BL/6J mouse (stranger 1), with no prior contact with the test mice, was enclosed in one of the wire cages. Next, test mice were returned to the middle chamber and allowed to explore for 10 min (sociability test). After the test session, the test mice were again placed in the holding cage and a second unfamiliar mouse (stranger 2) was enclosed in the wire cage on the opposite side. Test mice were placed in the middle chamber. They had a choice between the first, already investigated unfamiliar mouse and the novel, unfamiliar mouse for 10 min (social novelty preference test). The time spent in each chamber and around each cage was automatically measured from images using ImageCSI software (see “Section 4.5.14”).

#### 4.5.9. Acoustic Startle Response/Prepulse Inhibition Test

A startle reflex measurement system (O’Hara & Co.) was used to measure the startle response elicited by a loud stimulus (ASR) and the PPI of the startle response, as previously described [31]. Mice were placed in a plastic cylinder mounted on a platform with an accelerometer. Mice were left undisturbed for 10 min and then subjected to test trials consisting of 6 trial types: 2 startle stimulus-only trials and 4 PPI trials. White noise of 110 dB or 120 dB (40 ms) was used as the startle stimulus for all trial types. The prepulse stimulus was presented 100 ms before the onset of the startle stimulus, and its intensity was 74 dB or 78 dB (20 ms). Four combinations of prepulse and startle stimuli were used (74–110, 78–110, 74–120, and 78–120 dB). Six blocks of the six trial types were presented in a pseudorandom order, such that each trial type was presented once within a block. The average inter-trial interval was 15 s (10–20 s). Startle response was recorded for 400 ms, starting with the onset of the startle stimulus. The peak startle amplitude was used as the dependent variable. The background noise level was 70 dB during all test sessions. Percent PPI was calculated for each mouse, according to the following formula: percent PPI = 100 × [1 − (ASR amplitude in prepulse + startle trial)/(ASR amplitude in startle stimulus alone trial)].

#### 4.5.10. Porsolt Forced Swim Test

The Porsolt forced swim test, developed by Porsolt et al. [32], was performed to assess depression-related behavior. A Plexiglas cylinder (20 cm height × 10 cm diameter) was placed in a test chamber (49 cm height × 44 cm length × 32 cm width; O’Hara & Co.). A video camera was mounted on the ceiling of the test chamber and positioned directly above the cylinder. Mice were placed into a cylinder, and it was filled with water (approximately 23 °C) to a height of 7.5 cm. Immobility times were recorded over a 10 min test period on 2 consecutive days. Images were captured at two frames per second using a video camera and transferred to a computer. The area (pixels) within which the mouse moved was measured for each pair of successive frames. When the area was below a certain threshold, mouse behavior was judged as “immobile.” When the area equaled or exceeded the threshold, the mouse was considered “moving.” The optimal threshold for this judgment was determined by adjusting the threshold to the amount of immobility measured by a trained human observer. Immobility lasting <2 s was not included in the analysis. Data acquisition and analysis were performed automatically using ImagePS software (see “Section 4.5.14”).

#### 4.5.11. T-Maze Spontaneous Alternation Test

The spontaneous alternation task was conducted to assess spatial working memory using a modified automatic T-maze apparatus (O’Hara & Co.), as previously described [33,34]. The apparatus was constructed using white plastic runways with 25-cm-high walls. It was partitioned into six areas: the stem of the T, straight runway, left and right arms, and the connecting passageways from the arms to the stem of the T. Mice were subjected to a session with 10 trials per day for one day (cut-off time, 50 min). Each trial consisted of a forced choice, followed by a free choice (inter-trial interval, 60 s). In the forced-choice trial, mice were forced to enter either the left or right arm of the T-maze and were held in the arm for 10 s. After the 10 s period, the doors of the connecting passageway from the arm to the stem of the T were opened, and the mouse could return to the starting compartment. A free-choice trial was started three seconds after the mice entered the starting compartment. Mice were allowed to choose one of the arms. The percentage of correct responses in which the mice entered the arm opposite their choice in the forced-choice trial during the free-choice trial was calculated. Data acquisition and analysis were performed automatically using ImageTM software (see “Section 4.5.14”).

#### 4.5.12. Tail Suspension Test

The tail suspension test, developed by Stern et al. [35], was performed to assess depression-related behavior. Each mouse was suspended 30 cm above the floor by its tail in a white plastic chamber (44 cm height × 49 cm length × 32 cm width, inside dimensions; O’Hara & Co.), with a video camera mounted on the wall (O’Hara & Co.). The behavior was recorded for 10 min. Images were captured at two frames per second using a video camera. Similar to the Porsolt forced swim test, the immobility of each mouse was judged according to a certain threshold using ImagePS software.

#### 4.5.13. Contextual and Cued Fear Conditioning Test

Contextual and cued fear-conditioning tests were performed, as previously described [36]. In the conditioning session, each mouse was placed in an acrylic chamber consisting of white (side) and transparent (front, rear, and top) walls (33 × 25 × 28 cm) with a stainless-steel grid floor (0.2 cm diameter, spaced 0.5 cm apart; O’Hara & Co.). Mice were allowed to freely explore the chamber for 120 s, and 55 dB of white noise served as the conditioned stimulus (CS) for 30 s. During the last 2 s of CS presentation, a mild foot shock (0.3 mA, 2 s) was delivered as the unconditioned stimulus (US). Mice were subjected to two more CS-US pairings with a 2 min interstimulus interval. The animals were returned to their home cages 90 s after the last foot shock. Approximately 24 h after conditioning, the context test was conducted for 300 s. In the context test, mice were placed in the same chamber in which they had been conditioned. At least 2 h after the context test, a cued test with an altered context was performed for 360 s. In the cued test, the mice were placed in a triangular chamber (33 × 29 × 32 cm) made of white plastic walls and floor, which was located in a different sound-attenuating room, and allowed to explore the triangular chamber for 180 s. The CS was then presented during the last 180 s of the cued test. In each session, the percentage of freezing and the distance traveled were calculated automatically using ImageFZ software (see “Section 4.5.14”).

#### 4.5.14. Image Analysis

The application software used for behavioral experiments (ImageLD/EP/SI/CSI/PS/TM/FZ), which is based on the public domain ImageJ program (http://rsb.info.nih.gov/ij/, accessed on 1 January 2023), was developed and modified for each test by Tsuyoshi Miyakawa. The ImageLD/EP/TM/FZ programs can be freely downloaded from the “Mouse Phenotype Database” (http://www.mouse-phenotype.org/, accessed on 1 January 2023).

### 4.6. Real-Time Polymerase Chain Reaction (Real-Time PCR)

After euthanasia, we rapidly removed the whole brain using a 1 mm wide coronal section brain slicer (MUROMACHI, Tokyo, Japan), in which 1–2 mm sections posterior to the Bregma were cut, and the cortex, amygdala, hippocampus, and hypothalamus were sampled and RNA was extracted. Glyceraldehyde-3-phosphate dehydrogenase (GAPDH) was used as the internal control. All probes were purchased from Applied Biosystems, and amplification was performed using a LightCycler 96 System (Roche Diagnostics, Indianapolis, IN, USA). The cycling parameters were as follows: 50 °C for 120 s, 95 °C for 600 s, followed by 40 cycles of amplification at 95 °C for 15 s and 60 °C for 60 s.

### 4.7. Enzyme-Linked Immunosorbent Assay (ELISA)

Lipocalin-2 levels in the mouse plasma were measured using specific ELISA Kits (Abnova, Taipei, Taiwan). The experiments were performed in duplicate. The absorbance values were measured using a Multiskan FC (Thermo Fisher Scientific).

### 4.8. Culture of Astrocytes

Astrocytes were co-cultured with several cytokines to elucidate the direct effect of major inflammatory cytokines on the brain, and mRNA expression levels were measured. Cells were purchased from COSMO BIO Co., Ltd. (Tokyo, Japan), with astrocyte-specific medium (COSMO BIO Co., Ltd., Tokyo, Japan) in a 12-well culture plate (Coster, NY, USA), with or without 50 ng/mL of TNF-α, IL-1β, or IL-17A (BioLegend, SanDiego, CA, USA), and then incubated for 24 h (n = 6, each group). 

### 4.9. Statistical Analysis

Statistical analyses were performed using PRISM software version 9 (GraphPad, San Diego, CA, USA). Two-group comparisons were performed using the unpaired *t*-test, and more than three groups were analyzed using one-way ANOVA, followed by Tukey’s multiple comparison test. Differences were considered statistically significant at *p* < 0.05. * *p* < 0.05, ** *p* < 0.01, *** *p* < 0.001, **** *p* < 0.0001.

## Figures and Tables

**Figure 1 ijms-24-05942-f001:**
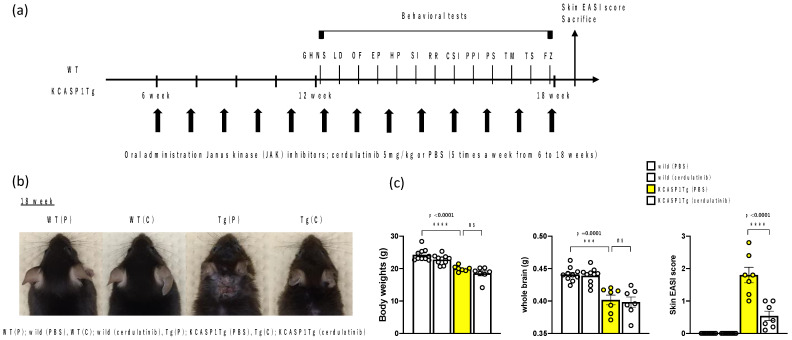
Behavioral experiments and JAK inhibitor treatment schedule; improvement of skin eruptions with JAK Inhibitor administration. (**a**) Schematic diagram of experimental procedures. (**b**) In 18-week-old KCASP1Tg mice, cerdulatinib treatment improved facial, head, and neck skin eruption. In WT mice, cerdulatinib treatment did not significantly alter skin symptoms compared with PBS treatment. (**c**) 18-week-old KCASP1Tg mice had lower body and whole brain weight compared to WT mice. Head and neck skin eruption of the mice was also evaluated using other sensory tests, using the EASI score, which is mainly used for the evaluation of atopic dermatitis (full score 7.2). The EASI score of KCASP1Tg mice significantly improved after cerdulatinib treatment. All data are expressed as the mean ± SEM by standard one-way ANOVA test, followed by Tukey’s multiple comparison test (*** *p* < 0.001, **** *p* < 0.0001). GHNS, general health and neurological screen; LD, light/dark transition test; OF, open field test; EP, elevated plus maze test; HP, hot plate test; SI, social interaction test; RR, rotarod test; CSI, three-chamber test; PPI, prepulse inhibition test; PS, Porsolt forced swim test; TM, T-maze spontaneous alternation test; TS, tail suspension test; FZ, contextual and cued fear conditioning test.

**Figure 2 ijms-24-05942-f002:**
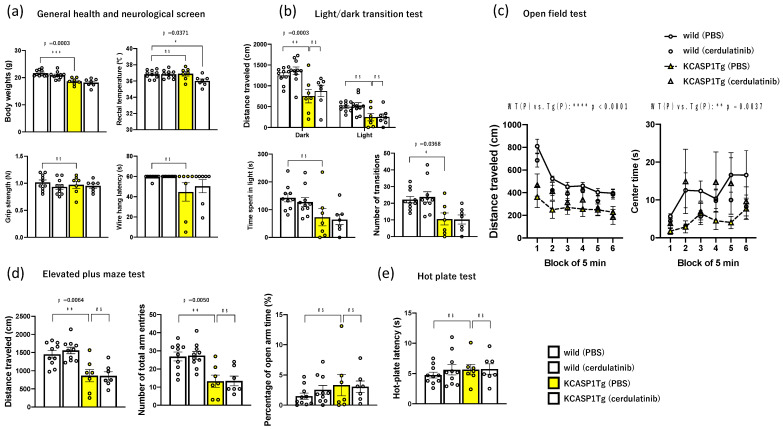
Behavioral analysis of four groups of KCASP1Tg and WT mice treated with PBS or cerdulatinib, respectively. (**a**) General health and neurological screen; KCASP1Tg mice had decreased body weights compared to WT mice. The grip strength of KCASP1Tg mice was not significantly different from that of WT mice. (**b**) Light/dark transition test; KCASP1Tg mice covered less distance in the dark chamber than WT mice and showed a decrease in the number of movements between the light and dark chambers, suggesting lower activity and higher anxiety-like behavior. Still, there was no predominant difference in the time spent in the light chamber. (**c**) Open field test; KCASP1Tg mice traveled less distance and spent less time in the center than WT mice, suggesting lower activity and higher anxiety tendencies. Cerdulatinib treatment did not improve distance traveled or center time in KCASP1Tg mice. (**d**) Elevated plus maze test; KCASP1Tg mice showed a decrease in distance traveled and the number of entries into the open and enclosed arms compared to WT mice, which implied low activity. There was no significant difference in the percentage of time spent in open arms. No behavioral changes were observed in KCASP1Tg mice after treatment with cerdulatinib. (**e**) Hot plate test; there was no difference in the predominance of pain sensitivity in the hot plate test between KCASP1Tg and WT mice. (**f**) Social interaction test; KCASP1Tg mice showed a decrease in distance traveled compared to WT mice, but there was no significant difference in the number of contacts or contact duration. (**g**) Rotarod test; KCASP1Tg mice showed no differences in coordinated locomotion or motor learning compared to WT mice. (**h**) Three-chamber test; KCASP1Tg mice spent less time in the cage containing stranger 1 than WT mice, indicating a decreased social behavior. In contrast, no difference was observed between KCASP1Tg and WT mice time spent around the cage containing stranger 1 and the time spent around the cage now containing a novel, unfamiliar mouse (stranger 2). (**i**) Acoustic startle response/prepulse inhibition test; KCASP1Tg mice showed reduced startle amplitude compared to WT mice, but there were no significant differences in prepulse inhibition compared to WT mice, suggesting a reduced response to acoustic stimuli. (**j**) Porsolt forced swim test; KCASP1Tg mice showed a decrease in immobility time on the second day compared to WT mice, suggesting that they may be more panicked. (**k**) T-maze spontaneous alternation test; KCASP1Tg mice took longer than WT mice but were not significantly different in correct responses or distance traveled. This suggests no deficit in working memory in KCASP1Tg mice. (**l**) Tail suspension test; KCASP1Tg mice were more active when suspended by the tail than WT mice. (**m**) Contextual and cued fear conditioning test; mice were exposed thrice to white noise as a conditioned stimulus (CS) for 30 s (horizontal black bars), followed by foot shock as an unconditioned stimulus (US) for the last 2 s of the conditioned stimulus (vertical arrows) during the conditioning session. Shock sensitivity is assessed based on the distance traveled during and after exposure to the unconditioned stimulus; there was no significant difference in immobility between KCASP1Tg and WT mice, suggesting no difference in contextual fear memory. All data are expressed as the mean ± SEM by ordinary one-way or two-way ANOVA, followed by Tukey’s multiple comparison test (* *p* < 0.05, ** *p* < 0.01, *** *p* < 0.001, **** *p* < 0.0001, ns: not significant).

**Figure 3 ijms-24-05942-f003:**
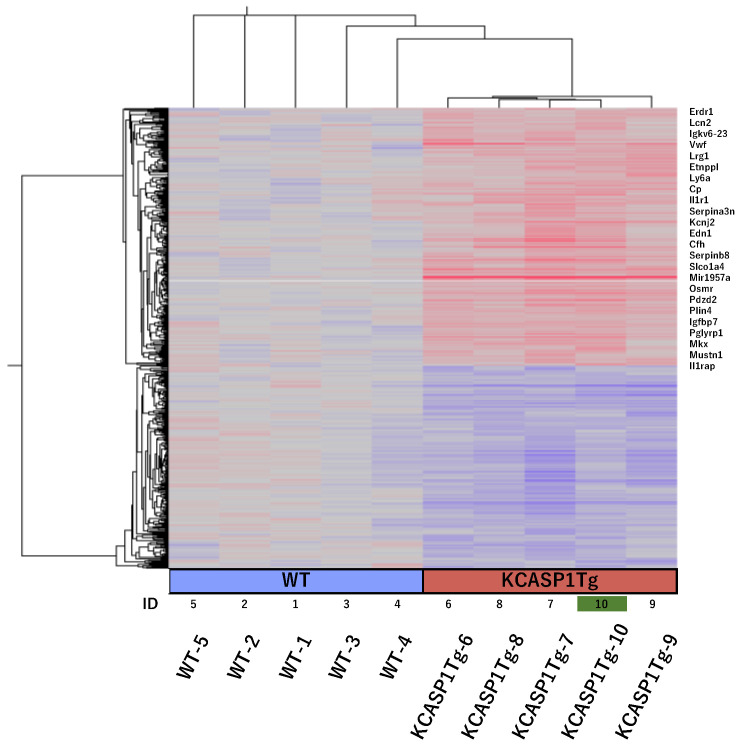
Microarray analysis of the cortex of 24-week-old KCASP1Tg and WT mice. Cerebral cortices of 24-week-old KCASP1Tg (*n* = 5) and WT (*n* = 5) mice were sampled for microarray analysis. *Erdr1* (erythroid differentiation regulator 1) and *Lcn2* (Neutrophil Gelatinase-associated Lipocalin; NGAL) gene expression was predominantly higher in KCASP1Tg mice compared to WT mice.

**Figure 4 ijms-24-05942-f004:**
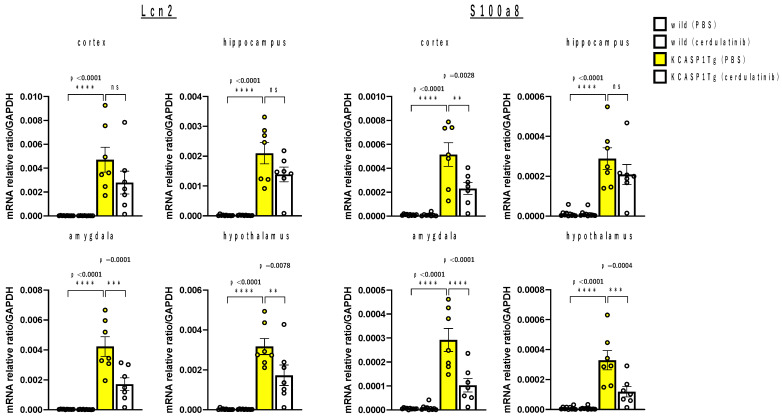
Lcn2 and S100a8 mRNA expression in the cortex, amygdala, hippocampus, and hypothalamus. After the completion of behavioral experiments, 18-week-old KCASP1Tg (*n* = 7) and WT (*n* = 10) mice were sacrificed, and each of the mRNA expressions of Lcn2 and S100a8, from four sites, were quantified using RT-PCR. Lcn2 and S100a8 were measured and standardized by the expression of GAPDH. KCASP1Tg mice had increased mRNA expression of Lcn2 and S100a8 in all regions of the cortex, amygdala, hippocampus, and hypothalamus compared to WT mice. In KCASP1Tg mice, cerdulatinib treatment decreased Lcn2 mRNA expression in the amygdala and hypothalamus, as well as S100a8 mRNA expression in the cortex, amygdala, and hypothalamus. All data were expressed as the mean ± SEM by standard one-way ANOVA test, followed by Tukey’s multiple comparison test (** *p* < 0.01, *** *p* < 0.001, **** *p* < 0.0001).

**Figure 5 ijms-24-05942-f005:**
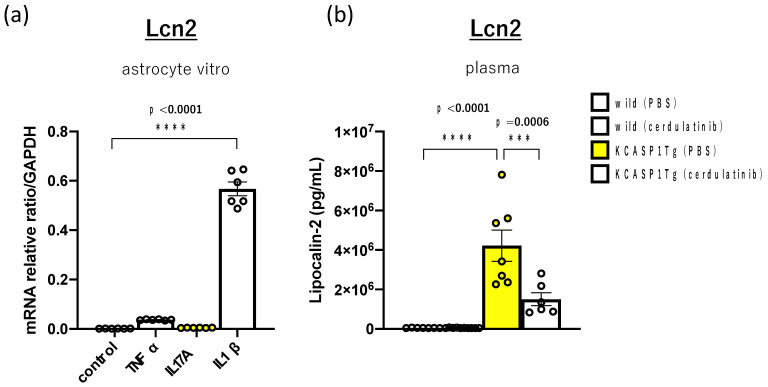
Lcn2 mRNA expression in astrocytes by inflammatory cytokine stimulation in vitro and Lcn2 in plasma. (**a**) We measured changes in mRNA expressions of the cultured astrocytes (*n* = 6) with inflammatory cytokines, including TNF-α, IL-17A, and IL-1β. Compared to other cytokines, stimulation with IL-1β increased Lcn2 mRNA expression in astrocytes. (**b**) At 18 weeks old, plasma Lcn2 was predominantly elevated in KCASP1Tg mice compared to WT mice. In KCASP1Tg mice, these decreased following cerdulatinib treatment. WT mice showed no significant difference after treatment with cerdulatinib. All data were expressed as the mean ± SEM by standard one-way ANOVA test, followed by Tukey’s multiple comparison test (*** *p* < 0.001, **** *p* < 0.0001).

## Data Availability

Not applicable.

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
