# Peer review of "Inflammatory Skin Disease Causes Anxiety Symptoms Leading to an Irreversible Course"

_ijms, 2023, doi:10.3390/ijms24065942_

Round 1

Reviewer 1 Report

Dear Authors,

in my opinion the paper is very interesting but the introduction can be improved with major information about the background on the argument. Actually in the introduction I reed information that could be in materials and methods.

Results are more rich and in my opinion are difficult to understand for the reader ; it is possible to simplify to make it suitable for all reader?

Thanks very much

Best regards

Author Response

Dear Authors,

in my opinion the paper is very interesting but the introduction can be improved with major information about the background on the argument. Actually in the introduction I reed information that could be in materials and methods.

Results are more rich and in my opinion are difficult to understand for the reader ; it is possible to simplify to make it suitable for all reader?

Thanks very much

Best regards

Response: Thank you for your suggestion. We have reduced the contents in the introduction session and moved to the materials and methods. We also made the results more simplified. We appreciate your comments.

Reviewer 2 Report

Comments to the Authors of Manuscript Number jcm-2197844

with the title

Inflammatory skin disease causes anxiety symptoms leading to an irreversible course

 This is an excellent research about behavioral changes in patients with inflammatory skin disease that are frequently encountered in current dermatological practice. I suggest that material and methods become section 2, the results – section 3 and discussion – section 4.

Author Response

 This is an excellent research about behavioral changes in patients with inflammatory skin disease that are frequently encountered in current dermatological practice. I suggest that material and methods become section 2, the results – section 3 and discussion – section 4.

Response: Thank you for your suggestion. According to the regulation, the construct of this journal should be in the following order.

  1. Introduction, 2. Results, 3. Discussion, and 4. Materials and Methods

Reviewer 3 Report

Authors applied the mouse model of keratin 14-driven caspase-1 overexpressing (KCASP1Tg) mice and cerdulatinib to explore relationship between the behavior change especially anxiety symptoms and skin inflammation. Although the role of skin inflammation in anxiety symptoms is an important issue in dermatological sciences, this study design and manuscript content can be further improved. Generally, the description of results is too simplified and cannot well explain the experimental findings. The content of figure legends is too much. The architecture of this manuscript should be modified. Some specific comments are listed.

1. In the introduction, authors mentioned the features of KCASP1Tg mice fulfilled the criteria of atopic dermatitis and could be used as the model for developing atopic dermatitis. However, I don’t think the profiles in this mouse model can mimic atopic dermatitis. Was there any marker of type 2 inflammation elevated in this model?

2. I wonder the reason why authors chose cerdulatinib, a Syk/Jak inhibitor but not other current Jak inhibitors which have been used for the treatment of atopic dermatitis.

3. Was Lcn2 the major upregulated gene in microarray analysis of the cortex of   KCASP1Tg compared to WT? The differential gene expression by microarray analysis mentioned in this study is quite limited. How about other genes?

4. I wonder the reason why authors chose antimicrobial peptides Lcn2 and s100a8 as the major targets explored in thus study. Was the result of s100a8 an incidental finding? Did authors also check changes of other markers associated with atopic dermatitis? 

5. Authors said the behavioral abnormalities in KCASP1Tg mice did not improve despite JAK inhibition but elevated plasma Lcn2 in KCASP1Tg mice could be decreased by cerdulatinib. Does that mean Lcn2 is not related to anxiety symptoms?

6. S100a8 and Lipocalin 2 expression in amygdala and hypothalamus regions were higher in KCASP1Tg mice but were reduced by cerdulatinib. However, the behavioral abnormalities in KCASP1Tg mice could not be affected by cerdulatinib.What does this mean?

Author Response

Authors applied the mouse model of keratin 14-driven caspase-1 overexpressing (KCASP1Tg) mice and cerdulatinib to explore relationship between the behavior change especially anxiety symptoms and skin inflammation. Although the role of skin inflammation in anxiety symptoms is an important issue in dermatological sciences, this study design and manuscript content can be further improved. Generally, the description of results is too simplified and cannot well explain the experimental findings. The content of figure legends is too much. The architecture of this manuscript should be modified. Some specific comments are listed. 

Response: Thank you for your suggestion. We have simplified the diagram and figure legend to provide a more detailed explanation of the results of the behavioral analysis.

  1. In the introduction, authors mentioned the features of KCASP1Tg mice fulfilled the criteria of atopic dermatitis and could be used as the model for developing atopic dermatitis. However, I don’t think the profiles in this mouse model can mimic atopic dermatitis. Was there any marker of type 2 inflammation elevated in this model? 

Response: Thank you for your suggestion. First of all, this mouse model of dermatitis reveals spontaneous dermatitis without any external triggers. About 8 weeks after birth, the mice started to show dermatitis around the eyes and neck. The lesions extended to the ears and legs and finally became generalized. Dermatitis started as erosive erythematous patches and moist crusted areas. After repeated re-epithelization and inflammation, the lesions progressed to chronic lichenified plaques. Almost all the mice developed cataracts, as is the case with AD. The frequency of scratching behavior increased remarkably with the onset of the skin lesions.

The skin before the onset of skin symptoms showed no histopathological changes. After onset, it presented remarkable epidermotropic cell infiltration, papillomatous proliferation and thickening of the epidermis, partial epidermal defects, crusts, and parakeratosis. These changes resembled those observed in the acute stage of human atopic dermatitis. The dermis showed an increase in the number of infiltrating CD4 T cells and a remarkable increase in toluidine blue-positive mast cells, similar to the findings in AD.

The serum histamine levels in the model mice increased according to the extension of the skin lesions. Serum histamine levels corresponded with an increase in mast cells in the skin. Serum IgE levels increased by the time of symptom onset, and showed remarkable elevation, correlating with the severity of the skin rash. IgG1 increased remarkably, however, increases in IgG2 and IgM were limited, suggesting the allergic nature of the observed changes.

The expression of cytokine mRNAs in the affected skin showed an elevation of IL-4 and IL-5, which are not detected in normal mice. The increase in IL-4 production was confirmed with an enzyme- linked immunosorbent assay (ELISA) for the culture condition medium of the lesions.

In addition, we measured the cytokine production in splenic cells after stimulation with anti-CD3 antibodies. We detected elevation of IL-3, IL-4, and IL-5, indicating a systemic Th2 shift of cytokines.

Our model satisfies all the essential elements of the internationally accepted diagnostic criteria by Hanifin-Rajka, i.e., (1) itching, (2) typical distribution of skin rash, (3) chronic recurrent dermatitis, and (4) family history. At least three of these major criteria are required. Our model satisfies 11 of 12 minor elements, including xeroderma, elevated serum IgE, early onset, dermatitis of the hands and feet, cheilitis, conjunctivitis, cataract, facial erythema, lesions around the eyes, and aggravation due to environmental changes. We could not detect environmental antigen-specific IgE in their sera.

This information has been supplemented in the text.

  1. I wonder the reason why authors chose cerdulatinib, a Syk/Jak inhibitor but not other current Jak inhibitors which have been used for the treatment of atopic dermatitis. 

Response: In our previous paper, “IL-17A Is the Critical Cytokine for Liver and Spleen Amyloidosis in Inflammatory Skin Disease (Int J Mol Sci 2022, 23, doi:10.3390/ijms-23105726. )”, KCASP1Tg mice were treated with baricitinib, a JAK1,2 inhibitor currently used in clinical practice for atopic dermatitis, and also with cerdulatinib, a pan JAK inhibitor. The results showed that baricitinib treatment improved the dermatitis in KCASP1Tg mice, but cerdulatinib treatment improved dermatitis more; therefore, cerdulatinib was administered in the present study.

  1. Was Lcn2 the major upregulated gene in microarray analysis of the cortex of  KCASP1Tg compared to WT? The differential gene expression by microarray analysis mentioned in this study is quite limited. How about other genes? 

Response: Thank you for your suggestion. Microarray analysis of the cortex revealed a limited number of genes that were upregulated in KCASP1Tg mice compared to WT mice, with Lcn2 and Erdr1 among those that showed more than 4-fold increased expression. RT-PCR was performed for both genes, but no significant difference was observed in Erdr1.

  1. I wonder the reason why authors chose antimicrobial peptides Lcn2 and s100a8 as the major targets explored in thus study. Was the result of s100a8 an incidental finding? Did authors also check changes of other markers associated with atopic dermatitis? 

Response: Thank you for your suggestion. Previously we have also performed a microarray analysis of skin from 6-month-old KCASP1Tg and WT mice. S100a8 and S100a9, members of the s100 protein family were predominantly upregulated in KCASP1Tg mice. Therefore we investigated these proteins in the cortex, amygdala, hippocampus, and hypothalamus of the brain in addition. Microarray analysis of the cortex showed a 1.57-fold advantage in the expression of S100a8 and a 1.96-fold advantage in the expression of S100a9 in 6-month-old KCASP1Tg mice compared to 6-month-old WT mice. The expression of atopic dermatitis-associated type 2 cytokines, IL-4, 13, TSLP, IL-33, and IL-25 was unchanged in microarray analysis.

  1. Authors said the behavioral abnormalities in KCASP1Tg mice did not improve despite JAK inhibition but elevated plasma Lcn2 in KCASP1Tg mice could be decreased by cerdulatinib. Does that mean Lcn2 is not related to anxiety symptoms?

Response: In KCASP1Tg mice, JAK inhibitor treatment improved dermatitis and plasma Lcn2 level, but various inflammatory cytokines such as dermatitis-derived type 2 circulating in plasma were less completely suppressed like in WT mice, and this might affect cells such as brain astrocytes. A relationship between anxiety symptoms and Lcn2 has been suggested in the current study, and we believe that its function is irreversible. The irreversible changes may have resulted in anxiety symptoms.

  1. S100a8 and Lipocalin 2 expression in amygdala and hypothalamus regions were higher in KCASP1Tg mice but were reduced by cerdulatinib. However, the behavioral abnormalities in KCASP1Tg mice could not be affected by cerdulatinib. What does this mean?

Response: Thank you for your suggestion, in KCASP1Tg mice, Lcn2 and S100a8 were up-regulated in the amygdala, which is mainly involved in emotional expression and emotional behavior, and in the hypothalamus, which is closely related to endocrine and autonomic function regulation; which were improved in amygdala and hypothalamus after cerdulatinib treatment, but the behavior might be irreversible.

Reviewer 4 Report

 The manuscript "Inflammatory skin disease causes anxiety symptoms leading to an irreversible course" presents authors results on important factors related with inflammatory skin disease and anxiety symptoms.

 In their research, authors conducted complex analysis and presented many and various examined factors/variables (gene expression, behavioral abnormalities, use of JAK inhibitor, etc.), which finally gave valuable statistical results on this subject. So, I consider the manuscript potentially very useful for current knowledge in the field of psychoneuroimmunology.

However, there are some suggestions.

  In the present form, authors talk about inflammatory skin disease, but it is needed to give more data on the specific skin disease/condition, if possible. Do authors think only to atopic dermatitis? They mentioned EASI which is used for atopic dermatitis. Or do you think dermatitis in general?

In addition, is this a pilot study?

 It would be useful to present (for clinicians) shortly basic characteristics of „a dermatitis model“ at least in Discussion section - especially because you give results/data obtained on this model.  Also, it is difficult to follow many examined variables/parameters and, consequently, manuscript text, and I suggest adding a presentation of examined parameters in a form of scheme or figure for better follow and interpretations of the text.

ABSTRACT: The study and results could be more clearly presented – it is somewhat confusing for the readers and their understanding of the results.

INTRODUCTION: I suggest mentioning more factors/variables that were examined in the research and mention the field of psychoneuroimmunology.

METHODS: After the title Methods, it would be useful to mention your analyzed factors and variables (e.g. in 2-3 sentences). Also, authors can write crucial steps in your analysis. Namely, there are many examined factors and it is somewhat difficult to follow the factors.

RESULTS: If possible, I suggest adding a table with the presentation of crucial results obtained by each study parameter in one table.

DISCUSSION The authors examined the factors involved in the field of psychoneuroimmunology and psychodermatology, so it is needed to mention some data on these fields, especially those related with their research. Maybe, the authors could add more information on potential use of their results for further practice.

  However, since I am a clinician and I can comment on the text as a clinician, I think that it would be useful to check the obtained data by an expert from the field of statistics.

Author Response

The manuscript "Inflammatory skin disease causes anxiety symptoms leading to an irreversible course" presents authors results on important factors related with inflammatory skin disease and anxiety symptoms.

 In their research, authors conducted complex analysis and presented many and various examined factors/variables (gene expression, behavioral abnormalities, use of JAK inhibitor, etc.), which finally gave valuable statistical results on this subject. So, I consider the manuscript potentially very useful for current knowledge in the field of psychoneuroimmunology.

However, there are some suggestions.

In the present form, authors talk about inflammatory skin disease, but it is needed to give more data on the specific skin disease/condition, if possible. Do authors think only to atopic dermatitis? They mentioned EASI which is used for atopic dermatitis. Or do you think dermatitis in general?

In addition, is this a pilot study?

Response: Thank you for your suggestion. Although this mouse model is a model of atopic dermatitis, we believe that similar brain conditions may occur in inflammatory skin diseases in general, including psoriasis. As pointed out by the reviewer, we have added more detailed dermatitis and mouse conditions into the text. We consider this a preliminary small-scale study to determine the feasibility of the study design. We also believe that we should study this analysis in Lcn2 knockout mice and other models.

 It would be useful to present (for clinicians) shortly basic characteristics of „a dermatitis model“ at least in Discussion section - especially because you give results/data obtained on this model. 

Response: Thank you for your suggestion. We have added the characteristics of transgenic mice in the discussion section.

Also, it is difficult to follow many examined variables/parameters and, consequently, manuscript text, and I suggest adding a presentation of examined parameters in a form of scheme or figure for better follow and interpretations of the text.

Response: Thank you for your suggestion. The results of the behavior analysis were divided into multiple figures to simplify the process.

ABSTRACT: The study and results could be more clearly presented – it is somewhat confusing for the readers and their understanding of the results.

Response: Thank you for your suggestion. The abstract and results have been revised for better clarity.

INTRODUCTION: I suggest mentioning more factors/variables that were examined in the research and mention the field of psychoneuroimmunology.

Response: Thank you for your suggestion. The introduction was changed to describe the relationship between depression and anxiety symptoms that may arise from skin eruptions.

METHODS: After the title Methods, it would be useful to mention your analyzed factors and variables (e.g. in 2-3 sentences). Also, authors can write crucial steps in your analysis. Namely, there are many examined factors and it is somewhat difficult to follow the factors.

Response: Thank you for your suggestion. The description of the behavioral experiments is somewhat lengthy, but we describe at the beginning of each sentence what items are being measured.

RESULTS: If possible, I suggest adding a table with the presentation of crucial results obtained by each study parameter in one table.

Response: Thank you for your suggestion. We have listed the particularly important data from the behavioral analysis in the Supplemental Figure.

DISCUSSION The authors examined the factors involved in the field of psychoneuroimmunology and psychodermatology, so it is needed to mention some data on these fields, especially those related with their research. Maybe, the authors could add more information on potential use of their results for further practice.

Response: Thank you for your suggestion. We have described the results of the behavioral analysis in more detail.

  However, since I am a clinician and I can comment on the text as a clinician, I think that it would be useful to check the obtained data by an expert from the field of statistics.

Response: Thank you for your suggestion. All behavior analysis data were evaluated by a behavior analysis specialist.

Reviewer 5 Report

1.In the abstract there is no word about Psoriasis but in introduction the authors discussed about Psoriasis

2.Page 1 line 39 the authors induced a spontaneous dermatitis,but no psoriasis or atopic dermatitis.Please clarify and discuss

3.Page 2 line 46 Please explain this -TNF-α, IL-17A, and IL-23, in the lesioned skin, which influences the  blood

4,Page 2 lines 60 -66 please clarify and decide about therm dermatitis ,psoriasis-theese entities are different

5.Page 2 lines 70-74 the authors speaks again about atopic dermatitis,not about psoriasis.Please clarify again

6.Page 8 lines 224-227.Please explain and refer how can influence for an example contact palmar eczema this association ,,high complication rate of coronary artery and cerebrovascular diseases, which are often  fatal,,

There is a confusion between therms of the skin pathology like Psoriasis,Atopic Dermatitis -which is Non associated with cardiovascular diseases-and Eczema /Dermatitis,ex contact dermatitis.Please clarify and rewrite

7.Reference 32 has no pages

8.Shoji has 6 references as autocitations 25,28,30,32,33,35

Author Response

1.In the abstract there is no word about Psoriasis but in introduction the authors discussed about Psoriasis

Response: Thank you for your suggestion. We have supplemented the explanation for psoriasis in the abstract section.

2.Page 1 line 39 the authors induced a spontaneous dermatitis,but no psoriasis or atopic dermatitis.Please clarify and discuss

Response: Thank you for your suggestion. First of all, this mouse model of dermatitis reveals spontaneous dermatitis without any external triggers. About 8 weeks after birth, the mice started to show dermatitis around the eyes and neck. The lesions extended to the ears and legs and finally became generalized. Dermatitis started as erosive erythematous patches and moist crusted areas. After repeated re-epithelization and inflammation, the lesions progressed to chronic lichenified plaques. Almost all the mice developed cataracts, as is the case with AD. The frequency of scratching behavior increased remarkably with the onset of the skin lesions.

The skin before the onset of skin symptoms showed no histopathological changes. After onset, it presented remarkable epidermotropic cell infiltration, papillomatous proliferation and thickening of the epidermis, partial epidermal defects, crusts, and parakeratosis. These changes resembled those observed in the acute stage of human atopic dermatitis. The dermis showed an increase in the number of infiltrating CD4 T cells and a remarkable increase in toluidine blue-positive mast cells, similar to the findings in AD.

The serum histamine levels in the model mice increased according to the extension of the skin lesions. Serum histamine levels corresponded with an increase in mast cells in the skin. Serum IgE levels increased by the time of symptom onset, and showed remarkable elevation, correlating with the severity of the skin rash. IgG1 increased remarkably, however, increases in IgG2 and IgM were limited, suggesting the allergic nature of the observed changes.

The expression of cytokine mRNAs in the affected skin showed an elevation of IL-4 and IL-5, which are not detected in normal mice. The increase in IL-4 production was confirmed with an enzyme- linked immunosorbent assay (ELISA) for the culture condition medium of the lesions.

In addition, we measured the cytokine production in splenic cells after stimulation with anti-CD3 antibodies. We detected elevation of IL-3, IL-4, and IL-5, indicating a systemic Th2 shift of cytokines.

Our model satisfies all the essential elements of the internationally accepted diagnostic criteria by Hanifin-Rajka, i.e., (1) itching, (2) typical distribution of skin rash, (3) chronic recurrent dermatitis, and (4) family history. At least three of these major criteria are required. Our model satisfies 11 of 12 minor elements, including xeroderma, elevated serum IgE, early onset, dermatitis of the hands and feet, cheilitis, conjunctivitis, cataract, facial erythema, lesions around the eyes, and aggravation due to environmental changes. We could not detect environmental antigen-specific IgE in their sera.

This information was added to the discussion.

3.Page 2 line 46 Please explain this -TNF-α, IL-17A, and IL-23, in the lesioned skin, which influences the  blood

Response: Thank you for your suggestion. Caspase-1 is overexpressed in the skin of KCASP1Tg mice, resulting in the activation of IL-1β and IL-18, which spontaneously causes dermatitis on the face from around 8 weeks of age and spreads to the whole body. In our previous paper, “IL-17A Is the Critical Cytokine for Liver and Spleen Amyloidosis in Inflammatory Skin Disease (Int J Mol Sci 2022, 23, doi:10.3390/ijms-23105726. )”, KCASP1Tg mice have elevated expression of cytokines such as TNF-α, IL-17A, and IL-23 in their skin, suggesting that these inflammatory cytokines derived from dermatitis are circulating in the bloodstream. This information has been supplemented in the text.

4,Page 2 lines 60 -66 please clarify and decide about therm dermatitis ,psoriasis-theese entities are different

Response: Thank you for your suggestion. As described above, this mouse model of chronic inflammatory dermatitis can be evaluated as type 2 atopic dermatitis. However, since psoriasis, another inflammatory skin disease, has been shown to have similar psychiatric clinical manifestations, we decided to include content on psoriasis as well.

5.Page 2 lines 70-74 the authors speaks again about atopic dermatitis,not about psoriasis.Please clarify again 

Response: Please refer to the answer to question 4. We have modified this part.

6.Page 8 lines 224-227.Please explain and refer how can influence for an example contact palmar eczema this association ,,high complication rate of coronary artery and cerebrovascular diseases, which are often  fatal,,

There is a confusion between therms of the skin pathology like Psoriasis,Atopic Dermatitis -which is Non associated with cardiovascular diseases-and Eczema /Dermatitis,ex contact dermatitis.Please clarify and rewrite

Response: Thank you for your suggestion. There have been increasing statistics showing that inflammatory skin diseases such as atopic dermatitis and psoriasis increase the risk of internal organ complications, especially cerebrovascular and cardiovascular events, which can be fatal. Atopic eczema and major cardiovascular outcomes: A systematic review and meta-analysis of population based studies. (J Allergy Clin Immunol 2019, 143, 1821-1829, doi:10.1016/j.jaci.2018.11.030.) and Cause-specific mortality in patients with severe psoriasis: a population-based cohort study in the U K. (Br J Dermatol. 2010;163(3):586-592.) It is well known that the incidence of cardiovascular events is high in atopic dermatitis and psoriasis. In our previous paper, "Inflammatory skin march": IL-1-mediated skin inflammation, atopic dermatitis, and psoriasis to cardiovascular events ( J Allergy Clin Immunol. 2015 Sep;136(3):823-4. doi: 10.1016/j.jaci.2015.06.009. Epub 2015 Jul 26. PMID: 26220527.), we have proposed that inflammatory skin march is the inflammatory cytokine-induced vascular lesion caused by endothelial damage in chronic inflammatory skin diseases such as atopic dermatitis and psoriasis, and that a similar condition may occur in simple eczema if not controlled.

7.Reference 32 has no pages

Response: Thank you for your suggestion. We have supplemented the page number.

8.Shoji has 6 references as autocitations 25,28,30,32,33,35

Response: Thank you for your suggestion. Prof Shoji is an expert in behavioral experiments, and since we used the same method in this case, we have added it to the bibliography.

Round 2

Reviewer 1 Report

Thanks very much for the paper improvement , it can be accept

Reviewer 3 Report

no further comment